# The impact of the COVID-19 pandemic on stress and coping in parents of children with Autism Spectrum Disorder

Lenka Knedlíková[1], Lenka Dědková[2], Senad Kolář[1], Katarína Česká[1], Martina Vyhnalová[1,3], Lucie Stroupková[1,3], Jana Pejčochová[1], Theiner Pavel[4], David Lacko[2], Ondřej Horák[1], Hana Ošlejšková[1], Pavlína Danhofer[1] *

**1** Department of Paediatric Neurology, Faculty of Medicine, University Hospital Brno, Masaryk University, Brno, Czech Republic, **2** Interdisciplinary Research Team on Internet and Society, Faculty of Social Studies, Masaryk University, Brno, Czech Republic, **3** Department of Psychology, Faculty of Arts, Masaryk University, Brno, Czech Republic, **4** Department of Psychiatry, Faculty of Medicine, Masaryk University, University Hospital Brno, Brno, Czech Republic

* danhofer.pavlina@fnbrno.cz

**Data Availability Statement:** All relevant data are within the manuscript and its Supporting Information files.

## Abstract

### Introduction

Autism Spectrum Disorder (ASD) is a neurodevelopmental disorder characterized by impairments in communication, social interaction, and repetitive behavior. The declaration of the COVID-19 pandemic in March 2020 resulted in significant changes in daily life due to restrictive measures. This period posed particular challenges for families with children living with autism, given the limitations in medical care and social services.

### Objective

This study aimed to understand how families with autistic children perceive stress during the pandemic and the coping strategies employed in unexpected situations.

### Method

A total of 44 families with children with ASD and 300 control families, including 44 control families in a matched subsample, were included in the study. To assess stress and parental coping with COVID-19-related stress, the Responses to Stress Questionnaire (Adult Self-Report RSQ–COVID-19) was utilized.

### Results

Caregivers of autistic children experienced significantly higher stress levels (p = .027, d = 0.479) during the pandemic, with notable stressors such as limited access to medical care and challenges associated with remote work. Despite expectations, coping strategy differences were not statistically significant.

**Funding:** This work was funded by Ministry of Health of Czech Republic (AZV NU22-D-130). The funders had no role in study design, data collection and analysis, decision to publish, or preparation of the manuscript.

**Competing interests:** The authors have declared that no competing interests exist.

## Conclusion

Families and supporters of children with autism naturally encounter various experiences and challenges stemming from their additional needs. Our study's results highlight an accentuation of stress during challenging situations. As these situations may recur in the future, there is a need to design and implement support plans for these families, appropriate intervention programs, and preparations for the utilization of telemedicine tools.

## Introduction

Autism Spectrum Disorder (ASD) is classified as a neurodevelopmental disorder characterized by impairments in communication, social interaction, and repetitive behavior [1]. The prevalence of this disorder has been steadily increasing. Recent data indicates that it affects one in 36 children under the age of eight, posing a significant health and socioeconomic concern across various pediatric disciplines [2].

On March 11, 2020, the World Health Organization (WHO) declared the novel coronavirus (COVID-19) outbreak a global pandemic [3]. The onset of the COVID-19 pandemic brought about significant changes in all aspects of daily life, creating an unprecedented situation that impacted people worldwide. Stringent restrictions were implemented, leading to social isolation, limited social interactions, school closures, business shutdowns, and disrupted daily activities. The world became an uncertain and unpredictable environment. Fear, anxiety, and concerns for one's life and the lives of loved ones contributed to significant stress [4]. The disruption of daily activities, restricted access to medical care and social services, has been particularly challenging for children living with autism and their families, as evidenced by numerous studies [5–8]. Social isolation, limited service availability, loss of institutional support, reduced access to healthcare with limited therapeutic options due to restrictions have posed new types of challenges for caregivers, including higher levels of stress compared to families with neurotypical children [9]. Caregivers often faced negative economic impacts of the pandemic [10].

The COVID-19 pandemic also significantly disrupted the continuity of diagnosing neurodevelopmental disorders. This was due to sudden and substantial limitations in non-urgent healthcare. For instance, restrictive measures and infection concerns led to a decrease in diagnostic hospitalizations at the Department of Paediatric Neurology, Faculty of Medicine of Masaryk University Brno and University Hospital Brno, Czech Republic, from 56 in the period January 2018—December 2019 (24 months) to 28 during the strictest anti-epidemic measures in the Czech Republic from March 2020—December 2021 (22 months) [11]. Early detection of neurodevelopmental disorders is crucial for a child's prognosis, enabling the timely initiation of necessary interventions. Late ASD diagnosis contributes to increased difficulties in nonverbal communication, adaptive behavior, and the severity of ASD compared to groups with earlier diagnoses [12]. The pandemic era facilitated the implementation of telemedicine principles. In response to delayed ASD diagnoses, several tools were developed to enable remote diagnostics using audiovisual (AV) transmission [13, 14].

Stress is a multidimensional construct, involving psychological and physiological reactivity in response to events perceived as threatening to existence [15]. The manner in which one responds to stress can have significant immediate and long-term effects. Model Compas et al. presents a robust theoretical framework in which stress responses and coping are delineated into three dimensions: an active approach to problems and emotions (primary control), the acceptance and reevaluation of thoughts (secondary control), and the avoidance or denial of

stressors (disengagement) [16, 17]. Caring for a child living with autism entails specific challenges for caregivers, including the cultivation of virtues such as patience, adaptability, professional competence, and ongoing adjustment to the child's unique requirements. While providing tailored educational support may pose difficulties due to the limited availability and financial constraints associated with specialized interventions, it is important to acknowledge the innate resilience and creativity often exhibited by autistic children, despite the inherent challenges of adapting to changes in their surroundings and routines. In light of these considerations, the daily care of a autistic child should be viewed not solely as a burden, but also as an opportunity for personal growth and understanding. Although external factors such as the COVID-19 pandemic undoubtedly exacerbate the inherent challenges of caregiving, they also underscore the significance of empathy and assistance for individuals with ASD and their caregivers alike [18, 19]. By fostering a supportive and inclusive environment, we can mitigate stressors and enhance the well-being of all involved parties.

Our retrospective study aims to comprehensively compare the experiences of caregivers of autistic children and a control group during the COVID-19 pandemic. One aspect involves understanding nuances in the perception of stress in both groups, with additional analysis of stress coping strategies in unexpected challenging situations, such as the pandemic period.

## Method

### Sampling and procedure

The sample of caregivers or parents with children diagnosed with ASD (ASD group/sample) was recruited through questionnaires distributed at the Department of Pediatric Neurology and the Department of Psychiatry of the Faculty of Medicine at Masaryk University and University Hospital Brno, Czech Republic. The recruitment also took place at the School for children with ASD in Brno, Czech Republic, as well as at the institutions *Za sklem* and *Paspoint* in Brno, Czech Republic. The final sample consisted of 44 parents with children aged 3–17 (M = 8.48, SD = 3.40, 88.6% boys).

To collect data from the control group (parents from the general population), we collaborated with the professional research agency Median. The agency utilized its pool of online panelists (approximately 30,000 panelists) for participant recruitment. Eligibility criteria included parents or caregivers living in a common household with children (aged 0–18). Exclusion criteria involved living with children having any psychiatric, neurodevelopmental, or other severe chronic diagnosis. Quotas were employed to ensure the sample represented Czech households with children in terms of socioeconomic status (highest achieved education of parents), region (based on the Nomenclature of Territorial Units for Statistics Level 3 [NUTS3]; see European Commission & Eurostat, 2020, and municipality size [20]. Additionally, we established quotas for children's age (balanced for each of the six categories: 0–2, 3–5, 6–8, 9–11, 12–14, 15–18) and gender (3.5 times more boys than girls). The full control sample comprised 300 parents with children aged 0–18 (M = 8.78, SD = 5.35, 75.7% boys).

Both samples substantially differed in some key variables that could impact their experiences during COVID-19 (some ages of children were not present in the ASD group and this sample also consisted of families with higher socioeconomic status). Thus, we created a subsample of parents from the control group that would match the clinical sample better (see Analysis for details). The matched control subsample included 44 parents with children aged 3–17 (M = 8.50, SD = 4.31, 88.6% boys).

For both samples, data were collected using an online questionnaire. Data collection took place from 14 November 2022 to 25 July 2023. All study participants were thoroughly informed about the research and confirmed this either by signing an informed consent form

(standard adult informed consent form in written form) or online before starting and completing the questionnaire (online survey consent form). The Ethics Committee of the University Hospital Brno, Czech Republic, and the Ethics Board of the Faculty of Medicine of Masaryk University, Brno, Czech Republic, approved this project in writing and agreed with the realization.

## Measures

**Perceived stress and control over stress.** To evaluate stress, the Responses to Stress Questionnaire (Adult Self-Report RSQ–COVID-19) was employed [21]. The original version comprises 14 items inquiring about the perceived stressfulness of various aspects of parents' lives (e.g., "Financial problems because of COVID-19," "Trouble getting medical care or mental health services because of COVID-19"), measured on a 4-point Likert scale (1 = not at all, to 4 = very). Due to a technical error, two items (k and l) were inadvertently omitted from the final survey. Specifically, these items were: "k: "Uncertainty about when COVID-19 will end or what will happen in the future.", "i: "Difficulty completing my work responsibilities remotely because of COVID-19.". We used confirmatory factor analysis (CFA) with robust maximum likelihood estimator (MLR) to check the structure of the scale, which showed inadequate fit ($\chi^2(54) = 251.086$, p < .001, CFI = .838, TLI = .802, RMSEA = .131 [.115, .147], SRMR = .066). The detailed inspection revealed several high residual covariances, and the fit was acceptable after allowing four pairs of items to covary ($\chi^2(50) = 135.613$, p < .001, CFI = .930, TLI = .908, RMSEA = .089 [.072, .107], SRMR = .050). The covariances were between three items that concerned restrictions related to social activities or events (item b, c, and d), and three items that concerned the COVID-19 infection and symptoms (items g, h, and i). Thus, the residual covariances made theoretical sense. The final scale had a good internal reliability ($\omega$ = .881) and we used factor score for the main analysis. The section also included an item focused on perceived control over stress ("How much control you generally think you had over these problems") assessed on a 4-point rating scale (1 = none, 4 = a lot).

**Coping with stress.** To assess parents' coping with COVID-19 related stress, the Responses to Stress Questionnaire (Adult Self-Report RSQ–COVID-19) was utilized [21]. The original version consists of 57 items examining how parents dealt with COVID-19 related stress (e.g., "When dealing with the stress of COVID-19, I felt sick to my stomach or got headaches," "When I was around other people I acted like COVID-19 had never happened"), measured on a 4-point Likert scale (1 = not at all, to 4 = a lot). The questionnaire identifies 19 subfactors for responses to stress, which are combined into five main coping scales (Primary Control Coping, Secondary Control Coping, Disengagement Coping, Involuntary Engagement Coping, Involuntary Disengagement Coping). Due to a technical error, four items (10, 37, 43, and 45) were inadvertently omitted from the final survey. Specifically, these items were: "10: I just have to get away from everything when I am dealing with the stress of COVID-19," "37: When I am faced with the stressful parts of COVID-19, right away I feel really. . .," "43: I keep my mind off stressful parts of COVID-19 by. . .," and "45: I do something to calm myself down when I'm dealing with the stress of COVID-19.". We adhered to the author's guidelines and computed the scores when at least two out of three items per subfactor were filled in by the respondents–despite the omitted items, this still allowed using all subfactors.

To check the structure of the data, we again run CFA with MLR. Because of the relatively smaller sample size and high number of items, we used the item-parceliazation procedure in line with the author's approach [22]. We first checked internal reliabilities for parcels and omitted parcels with $\omega$ < .500 (two parcels). The remaining 17 parcels were thus used (with $\omega$ ranging from .517 to .800). The model reflecting the original structure showed significant

shortcomings and high multicollinearity leading to convergence issues and the Heywood case, hence we adjusted the expected structure: we omitted second order structure for voluntary coping, and because of extremely high correlation between involuntary engagement and disengagement (r > .90), we combined these two factors into one, following the approach of Benson et al. [23] REF[ld2]. Furthermore, we control for response style via modeling common method bias [24]. The final scale thus consists of four factors (with following internal reliabilities (ω): primary control .647, secondary control .766, disengagement coping .722, involuntary coping .948). Similarly, to perceived stress, we use factor scores for the main analyses. All items were adapted from the English version to Czech using the TRAPD model [25].

**Open-ended questions.** The Adult Self-Report RSQ–COVID-19 questionnaire incorporates a section with open-ended responses, allowing parents to intricately describe the stressful situations and challenges they encountered during the pandemic in the context of caring for a child with ASD. This segment facilitated parents in providing closer commentary on the coping strategies they adopted to navigate the specific stressors. Consequently, it enabled the identification of the most challenging issues and nuances in the behavior and functioning of individual families.

## Variables used for creating matching sample

**Parental education.** Respondents were asked to denote the highest achieved education of both parents, and the higher education level was used to match our samples. The response options were (1) primary or unfinished primary, (2) secondary without school-leaving examination, (3) secondary with school-leaving examination, (4) higher vocational school, (5) university.

**Perceived financial security** was assessed by a single item ("How does your household manage with the overall monthly income?") with a 6-point rating scale (1 = With great difficulty, 6 = Very easily).

**Municipality size** was assessed with a single item with three categories (1 = less than 5,000 inhabitants, 2 = between 5,000 and 100,000, 3 = more than 100,000).

**Experience with COVID-19.** Parents were asked about the most severe course of COVID-19 among family members. The severity was assessed on a 5-item rating scale (1 = no one got sick, 2 = mild course (i.e., cough, elevated temperature), 3 = moderate course requiring hospitalization (i.e., hospitalization in a standard ward), 4 = severe course requiring hospitalization (i.e., in the intensive care unit), 5 = severe course followed by death).

**Children's age** (in years) and **gender** (male, female) were also included.

## Analysis

To compare the scores on the examined scales between the ASD and control group of matched parents, we conducted t-tests and utilized Cohen's d for estimating effect sizes. In assessing perceived stress, we compared both the total score (average across items) and individual items independently to provide a comprehensive understanding of differences within the samples. For the analysis of control with stress (a single item), we computed the non-parametric Mann-Whitney U test with the Glass rank-biserial correlation coefficient (r) for effect size.

To construct the matched control subsample–a subsample mirroring the distribution of key variables in the ASD sample–we employed 1:1 nearest neighbor matching without replacement, utilizing a propensity score estimated through probit regression. Prior to matching, participants from the full control sample with children of ages differing from those in the ASD sample were excluded (N = 75). Covariates, including children's age and gender, parental highest education, perceived financial security, COVID-19 experience, and municipality size, were considered.

**Table 1. Balance diagnostics.**

| | Clinical sample (n = 44) | Full control sample (n = 300) | | | Matched control sample (n = 44) | | |
|---|---|---|---|---|---|---|---|
| Variables | M | M | SMD | VR | M | SMD | VR |
| Distance | 0.239 | 0.149 | 0.698 | 1.808 | 0.232 | 0.056 | 1.104 |
| Child's age | 8.477 | 9.773 | -0.382 | 0.663 | 8.500 | -0.007 | 0.620 |
| Child's gender | 1.114 | 1.262 | -0.463 | 0.530 | 1.114 | 0.000 | 1.000 |
| Parents' education | 4.023 | 3.507 | 0.448 | 0.915 | 4.023 | 0.000 | 0.851 |
| Perceived financial security | 3.841 | 3.373 | 0.378 | 1.488 | 3.796 | 0.037 | 1.683 |
| COVID-19 experience | 2.341 | 2.076 | 0.299 | 1.844 | 2.296 | 0.051 | 1.162 |
| Municipality size | 1.864 | 1.747 | 0.159 | 0.911 | 1.886 | -0.031 | 0.877 |

Note: M = mean, SMD = standardized mean difference, VR = variance ratio.

The average treatment effect in the treated (ATT) served as the target estimand. This matching specification resulted in satisfactory balance, as illustrated in Table 1. Post-matching, all variance ratios for covariates were between 0.5 and 2, standardized mean differences for covariates were below 0.1, and those for squares and two-way interactions between covariates were below 0.15, indicating effective balance [26]. Consequently, the matched control subsample corresponds well to the ASD sample.

Given the utilization of a 1:1 approach, the matched control subsample consisted of 44 parents. The matching process was conducted using the *R* software and the *MatchIt* package [27, 28]. The analysis of CFAs, internal consistency and factor scores extractions were done in R, packages semTools and lavaan [29, 30].

## Results

Table 2 provides a summary of our test results, highlighting distinctions in how parents from the ASD and control groups perceived stress and coped during COVID-19. Concerning perceived stress, a small to medium effect (d = 0.479, p = .027) was observed in the total score, indicating that parents in the ASD sample experienced more stress (M = 0.451) compared to parents in the matched control subgroup (M = 0.001). When it comes to perceived control over the stressful situations, the samples did not differ (ASD: M = 3.128, SD = 0.656, control: M = 2.886, SD = 0.910, p = .339, Glass r = .115).

Regarding coping strategies, no significant differences were observed between the ASD sample and the matched control subgroup. Effect size for secondary control coping (d = 0.209) suggests a small effect size might exist in this dimension, however our sample did not have sufficient statistical power to support it.

We focused on the outcomes that pertain to families of children with ASD. Open-ended responses utilized by 24 out of 44 families, representing 54.5%, and were reasonably consistent.

**Table 2. T-test results for scales.**

| | ASD sample (n = 44) | | | Matched control sample (n = 44) | | | t (df) | p | Cohen's d |
|---|---|---|---|---|---|---|---|---|---|
| | Range (min; max) | M | SD | Range (min; max) | M | SD | | | |
| Total Parent Perceived Stress | (-1.153; 2.525) | 0.451 | 0.889 | (-1.275; 3.054) | 0.001 | 0.991 | 2.246(86) | .027 | 0.479 |
| Primary Control Coping | (-1.564; 2.645) | 0.207 | 0.926 | (-1.874; 2.544) | 0.041 | 0.964 | 0.822(86) | .413 | 0.175 |
| Secondary Control Coping | (-2.001; 1.411) | 0.137 | 0.771 | (-2.136; 1.289) | -0.038 | 0.896 | 0.982(86) | .329 | 0.209 |
| Disengagement Coping | (-1.705; 1.779) | 0.109 | 0.901 | (-2.185; 2.639) | 0.108 | 0.972 | 0.005(86) | .996 | 0.001 |
| Involuntary Coping | (-1.581; 2.481) | 0.162 | 1.028 | (-1.537; 3.131) | 0.109 | 0.970 | 0.25(86) | .803 | 0.053 |

The responses enabled us to delineate three areas that presented the greatest stress burden for parents of children with ASD, namely: *unavailability of medical care and mental health services*, *challenges associated with remote work*, *increased family or work responsibilities*, and *implemented protective anti-pandemic measures*.

Parents from the ASD group reported higher stress related to difficulties in accessing medical care due to COVID-19 (M = 2.61, SD = 1.039) compared to parents from the matched control group (M = 2.02, SD = 1.067). Although the difference was not statistically significant due to the small sample size (t(86) = 2.631, p = .010), Cohen's d (.561) indicates a medium effect size difference, bolstering the assumption that the unavailability of medical care was especially distressing for parents with children with ASD.

## Discussion

### Introduction

The aim of the study was to assess the caregiver stress burden of autistic compared to the control group. We also focused on the coping strategies of both groups, all within the context of significant restrictions during the COVID-19 pandemic.

### Caregivers' perception of stress

We identified differences in stress perception between the ASD and control groups, where stress burden was higher compared to families with neurotypical children (Table 2).

Caregivers of children living with autism experience higher levels of stress, perceived depression, and anxiety among caregivers, considering the specificities of caring for a child with additional needs [18, 19]. This fact may be accentuated during unexpected stressful events, such as a pandemic [31]. Results indicate that families with neurotypical children also experienced stress burden. A new set of stressors threatening health, safety, and economic well-being during the COVID-19 pandemic was a global phenomenon [32]. The impact of the pandemic on all families could thus be perceived similarly, as we faced common challenges and lifestyle changes due to measures to curb the virus spread.

The course of the pandemic had a significant negative impact on the stress level of families with autistic children. A detailed examination of specific stressors from open-ended questionnaire responses brought interesting insights. Respondents independently reiterated several prominent aspects causing stress to supporters: *unavailability of medical care and mental health services*, *challenges associated with remote work*, *increased family or work responsibilities*, and *barrier measures* those autistic children found particularly limiting and stressful (especially wearing masks and hand sanitizing). This serves as a good illustration of the major challenges faced by families with autistic children during the pandemic. Therefore, we delve into specific stressors in more detail, even though statistically significant differences were not found in individual items. Insignificant results when comparing the ASD sample with the corresponding subsample can be attributed to the small sample size and, consequently, lower statistical power.

**Unavailability of care and mental health services.** The pandemic led to a sudden surge in testing and intensive care demands, overwhelming healthcare systems. This resulted in concerns about excluding vulnerable groups, including individuals with autism, from these essential services [33]. Limited physical access to specialized healthcare or therapeutic centers delayed the diagnosis and reduced the effectiveness of necessary interventions for autistic children [34]. In the U.S., 36% of parents of autistic children reported their children losing access to healthcare services. The findings of our study suggest that parents of children with autism experienced more stress related to difficulties in accessing medical care due to the COVID-19

pandemic compared to the control group of families with neurotypical children. Furthermore, upon examining open-ended responses, parents highlighted issues such as postponed diagnostic hospitalizations due to the pandemic, as well as the necessity of wearing masks, which limited the effectiveness of psychological assessments. Lastly, parents of autistic children reported difficulties in accessing respite care and residential facilities, which they considered stressful not only for themselves but also for their autistic child. These facilities represent essential environments for some children, where they interact with peers with similar diagnoses, making them invaluable for the children's well-being. Parents found telemedicine to be a useful alternative when available [35]. Recent developments in telemedicine diagnostic tools, utilizing modern technologies such as remote video meetings and audiovisual recordings, have garnered considerable attention with very positive results, though they are still not widely adopted in practice. Our recent study shows that experienced clinicians can formulate sufficiently accurate diagnostic conclusions regarding ASD using an online diagnostic tool Brief Observation of Symptoms of Autism (BOSA) in conjunction with the well-established tool The Autism Diagnostic Interview, Revised (ADI-R) [14].

**Challenges associated with remote work and increased family or work responsibilities.** In line with Corbett et al., we demonstrated that parents of children living with autism experience increased stress related to concerns about the future and challenges of remote work [36]. Work changes due to physical distancing measures caused massive job losses and economic uncertainty [37]. These changes contributed to a significant increase in parental burden, manifested by a high prevalence of depression (62.5%) and anxiety (20.5%) [9]. In relation to the severity of autistic features or behavior, parents had less time for themselves [38]. Providing care for someone with autism brings joys but also significant challenges, often demanding substantial commitments from caregivers. While prioritizing the well-being of their loved one, caregivers may inadvertently neglect their own health and relationships. However, recognizing the importance of self-care is essential for providing optimal care for the individual with autism.

**The impact of protective measures on children with ASD according to caregivers.** Problems with tolerating face masks and other protective measures were the most frequently mentioned issue in the open-ended responses of caregivers for autistic children. Autistic persons have their specific needs and perceptions, which sometimes require special considerations. For instance, they typically struggle with interpreting facial expressions and gestures, finding it less intuitive than neurotypical individuals. They often focus more on the mouth area [39–41]. Wearing masks thus had a negative impact on the social cognition of individuals with autism [42]. Sensory hypersensitivity is another typical trait of ASD; therefore, tactile sensations associated with wearing face masks were particularly uncomfortable for them [42].

## Coping strategies for stress management

In our evaluation, we relied on the model of coping strategies. Coping strategies delineate the ways in which individuals address problems and stressful situations. During the analysis of coping strategies, no statistically significant differences were discerned between the sample of parents with autistic children and the corresponding control group. The effect size for secondary control coping suggests that there might be a small effect in this domain; however, our dataset was not sufficiently large to confirm this statistically. Contrary to our results, findings from other studies indicate that parents of autistic children experience significantly higher levels of stress and symptoms of depression, often resorting to maladaptive coping strategies compared to parents of neurotypical children [40–42]. Our findings suggest that defining coping strategies used by parents of autistic children is challenging. The interpretation of our findings

indicates that parents raising an autistic child have the ability to cope with stressful situations in a broader context. They are accustomed to a higher burden and are motivated to develop functional coping mechanisms for long-term resolution of challenging situations. The ongoing pandemic adds another layer of stress, yet parents are capable of accepting and managing this stress through secondary control coping. Nevertheless, this is a hypothesis, and our findings must be interpreted cautiously. It is important to acknowledge that life with an autistic child may be demanding, but it also brings many positives and unique experiences.

## Study implications

From a methodological standpoint, our study also points out that the original structure of the used coping scale may not hold neatly nowadays. In particular, both dimensions of involuntary coping were highly correlated ($> .90$), and similarly high correlations were found in other studies, too [23, 43]. Such high correlations suggest that both dimensions almost completely overlap and that separating involuntary engagement and disengagement into two dimensions is not supported by the empirical data. Unfortunately, studies that used the scale to assess COVID-19-related coping rarely utilized any factor analysis techniques [36]. We thus encourage future researchers to be mindful of the need to test dimensional structure and in particular, to consider correlations between the originally proposed dimensions, which may require the respecification of the originally proposed structure.

## Study limitations

A strength of the study is the representative sample of parents and caregivers of children with ASD, including children with mild (N = 17), moderate (N = 19), and severe symptoms (N = 8). Another strength is the robust control sample, especially the matched control subsample, created very representative to the ASD group, allowing for a precise comparison of both groups. Unfortunately, its creation led to a significant reduction in the number of control participants in the study, bringing certain limitations. This is a retrospective study, which may present limitations such as difficulties in accurately recalling past events related to the pandemic, influencing the reliability of retrospective information. Due to an administrative error in distributing questionnaires, some questionnaire items were excluded, but this was accounted for in the methodology and did not affect the overall results. Evaluating the perception of stress and coping mechanisms in children with ASD would be interesting. However, considering the representation of younger age groups and the predominant number of children with moderate to severe ASD symptoms, obtaining these data validly was not feasible. While there are many limitations in such a study, the experiences reflected in the findings underscore the notable impact of the pandemic on children and parents with and without ASD.

## Conclusion

This study examined the impact of the COVID-19 pandemic on stress and coping mechanisms among parents of autistic children compared to families without children exhibiting autistic features. Our findings provide valuable insights into stress perception and coping mechanisms within families of autistic children, offering a foundation for future interventions and appropriate care during similar crises. Given the anticipated widespread mental health repercussions of the pandemic, our study underscores the necessity to bolster and implement supportive frameworks for these families, suitable intervention programs, and readiness to utilize tele-medicine tools. Instead of solely focusing on challenges, it is crucial to acknowledge the unique

strengths and perspectives of individuals with autism. Through fostering empathy and providing resources, we can create a more inclusive and supportive environment for all.

## Supporting information

**S1 Data.**
(SAV)

## Author Contributions

**Conceptualization:** Lenka Knedlíková, Pavlína Danhofer.

**Data curation:** Lenka Dědková, Katarína Česká, Martina Vyhnalová, Lucie Stroupková, Jana Pejčochová, Theiner Pavel, David Lacko, Ondřej Horák.

**Formal analysis:** Lenka Dědková, David Lacko.

**Investigation:** Lenka Knedlíková, Katarína Česká, Martina Vyhnalová, Lucie Stroupková, Jana Pejčochová, Theiner Pavel, Ondřej Horák.

**Methodology:** Lenka Dědková, David Lacko.

**Project administration:** Senad Kolář.

**Software:** Senad Kolář.

**Supervision:** Hana Ošlejšková, Pavlína Danhofer.

**Validation:** Lenka Dědková.

**Writing – original draft:** Lenka Knedlíková, Pavlína Danhofer.

**Writing – review & editing:** Lenka Knedlíková, Pavlína Danhofer.

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
