## [Decision Letter · Decision Letter 0]

26 Mar 2024

PONE-D-24-03555The impact of the COVID-19 pandemic on stress and coping in parents of children with autism spectrum disorder.PLOS ONE

Dear Dr. Danhofer,

Thank you for submitting your manuscript to PLOS ONE. After careful consideration, we feel that it has merit but does not fully meet PLOS ONE’s publication criteria as it currently stands. Therefore, we invite you to submit a revised version of the manuscript that addresses the points raised during the review process.

We look forward to receiving your revised manuscript.

Kind regards,

Wen-Jun Tu

Academic Editor

PLOS ONE

“This work was funded by Ministry of Health of Czech Republic (AZV NU22-D-130).”

“This work was funded by Ministry of Health of Czech Republic (AZV NU22-D-130).”

“This work was funded by Ministry of Health of Czech Republic (AZV NU22-D-130).”

Reviewers' comments:

Reviewer's Responses to Questions

**Comments to the Author**

1. Is the manuscript technically sound, and do the data support the conclusions?

Reviewer #1: Yes

2. Has the statistical analysis been performed appropriately and rigorously? 

Reviewer #1: Yes

3. Have the authors made all data underlying the findings in their manuscript fully available?

Reviewer #1: Yes

4. Is the manuscript presented in an intelligible fashion and written in standard English?

Reviewer #1: Yes

5. Review Comments to the Author

Reviewer #1: The authors mentioned that due to technical errors, certain items were omitted from the final survey, specifically items (k and l) and items (10, 37, 43, and 45). I would suggest including in the main text a description of what these items were to provide readers a complete understanding of the survey content.

The section discussing the unavailability of care and mental health services is crucial. Consider drawing a clearer connection between this discussion and the specific objectives and findings of the study.

There appears to be an incomplete sentence from lines 321 to 323.

In the study limitations section, the authors mentioned information about the level of support of autistic children. If you wish to share more detailed information, consider placing it in the Results section or referencing it if the information is from supplementary material.

Additionally, I strongly recommend a thorough revision of the text to adopt a more neurodiversity and respectful tone when discussing autism. It is crucial to approach the topic in a manner that does not stigmatize or portray autistic children as a burden. The language used throughout the manuscript should reflect an understanding and supportive perspective toward autism, emphasizing the strengths and unique qualities of autistic individuals rather than focusing solely on challenges or negative aspects. This shift in narrative can contribute significantly to reducing societal stigma and promoting a more inclusive and empathetic understanding of autism. Please consider reviewing the entire manuscript to ensure that the language and framing consistently align with these principles.

6. PLOS authors have the option to publish the peer review history of their article (what does this mean?). If published, this will include your full peer review and any attached files.

Reviewer #1: **Yes: **Michele Bolan

---

## [Author Response · Author response to Decision Letter 0]

26 Apr 2024

Dear Editor and Reviewers, dear Dr. Bolan,

We sincerely appreciate your insightful and valuable comments on our research article, "The Impact of the COVID-19 Pandemic on Stress and Coping in Parents of Children with Autism Spectrum Disorder," and for the opportunity to submit a revised manuscript.

Your feedback is highly valuable to us and has helped us enhance the quality of our work. We have made every effort to address your constructive comments and suggestions comprehensively.

Regarding the journal requirements:

1) We have reviewed the stylistic templates provided by PLOS ONE. We believe that the research article meets the stylistic requirements of the journal, including those concerning file naming.

2-3) We have revised the wording of the Financial Disclosure as requested. The current wording is as follows: "This work was funded by the Ministry of Health of the Czech Republic (AZV NU22-D-130). The funders had no role in study design, data collection and analysis, decision to publish, or preparation of the manuscript." Additionally, as per your instructions, we have removed any text related to funding from the manuscript. Kindly incorporate this change in the online submission form, as suggested by yourselves.

4) The full data for the study has been added to the online form in the Supporting information files section. It was not possible to include open-ended responses in the published dataset, as they contain identifying information about the subjects and do not include variables that are part of the main analysis.

5) We have added the full names of both ethics committees and the method of obtaining their consent to the relevant section in the Methods section.

Reviewers' comments section:

1) To ensure that readers fully understand the survey content, we have added the full wording of items (k and l) and items (10, 37, 43, and 45) to the Measures section, which were omitted from the final survey due to technical errors.

2) We have reworked the paragraph on Unavailability of care and mental health services as recommended. We have provided more detailed results regarding the stress associated with the unavailability of healthcare and mentioned them both in the Results section and in the relevant paragraph of the Discussion. This better connects the results of our study with the discussion and expands upon them with specific parental concerns from the open-ended responses.

3) We have corrected and completed the inadvertently unfinished sentence between lines 321 to 323.

4) We have chosen to retain the Study Limitations section in its unchanged form.

5) Thank you for your comment on the topic of neurodiversity and the corresponding tone of the entire text. We have decided to fully respect this principle; therefore, we have substantially reworked the entire article in line with the Neurodiversity Paradigm advocated by WHO. For example, we replace inappropriate terms like "Child with ASD" with "Autistic child/child with autistic features," "Healthy children" with "Neurotypical children" etc. Instead of stigmatization, we aim to promote awareness, understanding, and acceptance of diversity.

Thank you once again for your thorough review and guidance throughout this process. We hope our revised manuscript will now be acceptable for publication. If you have any further suggestions for revisions, we would be happy to address them.

Yours sincerely,

Pavlina Danhofer and other members of the research team

---

## [Editor Report · Decision Letter 1]

24 Oct 2024

The impact of the COVID-19 pandemic on stress and coping in parents of children with autism spectrum disorder.

PONE-D-24-03555R1

Dear Dr. Danhofer,

We’re pleased to inform you that your manuscript has been judged scientifically suitable for publication and will be formally accepted for publication once it meets all outstanding technical requirements.

Kind regards,

Wen-Jun Tu

Academic Editor

PLOS ONE
---

## [Editor Report · Acceptance letter]

29 Oct 2024

PONE-D-24-03555R1 

PLOS ONE

Dear Dr. Danhofer, 

I'm pleased to inform you that your manuscript has been deemed suitable for publication in PLOS ONE. Congratulations! Your manuscript is now being handed over to our production team.

Kind regards, 

on behalf of

Dr. Wen-Jun Tu 

Academic Editor

PLOS ONE